# Egg Cooling After Oviposition Extends the Permissive Period for Microinjection-Mediated Genome Modification in *Bombyx mori*

**DOI:** 10.3390/ijms252312642

**Published:** 2024-11-25

**Authors:** Keiro Uchino, Ryusei Waizumi, Megumi Sumitani, Hiroki Sakai, Nobuto Yamada, Katsura Kojima, Naoyuki Yonemura, Ken-Ichiro Tatematsu, Tetsuya Iizuka, Hideki Sezutsu, Toshiki Tamura

**Affiliations:** 1Institute of Agrobiological Sciences, National Agriculture and Food Research Organization, 1-2 Owashi, Tsukuba 305-8634, Ibaraki, Japan; waizumir957@naro.affrc.go.jp (R.W.); sumikasashima@naro.affrc.go.jp (M.S.); sakaih786@naro.affrc.go.jp (H.S.); yamadan890@naro.affrc.go.jp (N.Y.); kojikei@naro.affrc.go.jp (K.K.); yonemura@naro.affrc.go.jp (N.Y.); k.tatematsu@naro.affrc.go.jp (K.-I.T.); tiizuka@affrc.go.jp (T.I.); hsezutsu@naro.affrc.go.jp (H.S.); 2Independent Researcher, Tsukuba 300-1207, Ibaraki, Japan; monsan20@yahoo.co.jp

**Keywords:** microinjection, permissive period for microinjection, developmental threshold temperature, *Bombyx mori*

## Abstract

In general, transgenesis efficiency is largely dependent on the developmental status of eggs for microinjection. We investigated whether the relationship between transgenesis efficiency and cooling eggs in silkworms, *Bombyx mori*, affects the transgenesis frequencies. First, we performed a microinjection using eggs of different developmental statuses at 25 °C. As a result, the use of eggs at 4 h after egg-laying (hAEL) demonstrated nearly five times greater efficiency in frequency compared to 8 hAEL but no transgenesis was found at 12 hAEL. Second, we examined the use of eggs stored for 5 or 24 h at 10 °C. We found that transgenic silkworms were produced not only 5 hAEL but also 24 hAEL. Finally, in the *BmBLOS2* gene knock-out experiment, eggs stored at 10 °C demonstrated knock-out phenotypes even 48 hAEL at the time of injection (G_0_). These results demonstrate that an egg cooling treatment enables drastically enhanced rates of efficiency for insect genome modification. Our results could be useful in other insects, especially species with an extremely short syncytial preblastodermal stage.

## 1. Introduction

In post-genomics, transgenesis is very useful for various basic studies and practical applications, including new crop and animal cultivation and breeding systems, gene therapy, and biological pest control [1,2,3,4,5,6,7,8,9,10,11,12]. In *Bombyx mori* [13], Tamura et al. were the first to succeed in producing transgenic silkworms [14]. To date, transgenic silkworms have been used not only for academic research, such as biology, genetics, physiology, neuroscience, and pathology [15,16,17,18,19,20,21,22,23,24,25,26,27] but also for industrial applications and as useful materials for therapeutic products [28,29,30,31,32,33,34,35,36,37,38,39]. Microinjection is a representative tool used genome modification across many kinds of animals, including arthropods [40,41,42,43,44,45,46,47,48]. In addition to microinjection, there are several methods which can be used to produce transgenic animals, including electroporation [49,50], baculovirus-mediated gene targeting [51], DNA-coated microparticle bombardment (biolistic) [46,52,53], sperm-mediated gene transfer [54], and the receptor-mediated ovary transduction of cargo [55]. Nevertheless, microinjection remains useful for certain embryonic experiments as it allows for the direct introduction of not only genes (DNA/RNA), but also a variety of sources such as proteins, chemicals, and even intercellular nuclei into eggs. This technique thereby enables researchers to examine the function of these materials at the moment of injection (G_0_). For successful microinjection, there are several factors that must be considered, such as the preparation of materials, skill of operators, development of the injection system, and so on. In particular, it is very important for the microinjection to be completed during the syncytial preblastodermal stage, the permissive period for successful genome modification. If not performed during this stage, the syncytial nuclei will eventually be enveloped by membranes, preventing the transformation event from occurring in the silkworm. This permissive window is regarded as being within 8 h but may be significantly shorter in animals with fast development stages such as *Drosophila*, *Caenorhabditis*, *Danioa*, and *Anopheles* [45,46,47]. Given the demand for high transgenesis efficiency, the syncytial preblastodermal stage injection is essential. Especially, in the case of *Bombyx mori*, there is a special situation where the microinjection requires two steps due to their hard eggshell. The first step involves making a hole with a tungsten needle, and the second step involves the precise insertion of a glass capillary into that hole, which may take a long time. Additionally, their transgenesis efficiency is approximately 2~30% in G_1_, which is lower than other animals [14,45,46,47,56], so it is necessary to inject approximately 700~1000 eggs per gene, which is quite time-consuming.

Here, we considered whether it was possible to sustainably maintain suitable conditions for embryonic microinjection to produce transgenic silkworms, even when some time has passed since egg laying. This study was performed in reference to the report by Sudo et al. [57], who found that there was a boundary of growth at 11 °C (threshold temperature) in the larvae of the silkworm, *Bombyx mori*. As a result, we could successfully extend the permissive period for transgenesis after oviposition to 24 h by storing eggs at 10 °C. Moreover, through knock-out experiments of *BmBLOS2*, the *B. mori* homolog of the human biogenesis of lysosome-related organelles complex1, subunit 2, which shows an oily mutant phenotype in the larval skin during the G_0_ phase, we found that cooling the eggs was effective even at 48 hAEL at 10 °C, and in the case of storage at −3~10 °C. Our findings in this study are applicable to insects with a short syncytial preblastodermal stage and strongly support novice researchers in microinjection.

## 2. Results

### 2.1. Transgenesis Efficiency by Microinjection Using Eggs in Different Developmental Stages

To examine the relationship between the elapsed time of eggs and transgenesis efficiency in microinjection, we performed microinjection using eggs that were pasted for 4, 8, and 12 h after the egg-laying (hAEL) stages. Detection of the transgenic silkworm in G_1_ was performed using the *3xP3-DsRed2* gene, which is expressed in the eye but, in some cases, the central nervous system, too, as shown in Figure 1. As shown in Table 1, the hatching proportions were almost the same between the 4 and 8 hAEL, but in low at the 12 hAEL: 53.6% ± 7.6%, 55.7% ± 2.8%, and 38.0% ± 6.0% in Mean ± SE, respectively. However, there was no significance found between these: a Kruskal–Wallis test, t = 3.2889, *p* = 0.1931. After microinjection, the grown moths were crossed with each other (sibling mating), or the remaining moths were crossed with the host strain moths: pnd-w1. As a result, the transgenesis efficiency at the 4 hAEL was much higher than at the 8 hAEL, but this effect was not found at the 12 hAEL: 28.6% ± 4.8%, 1.7% ± 1.2%, and 0.0% ± 0.0% in Mean ± SE, respectively (Table 1). Based on the statistical analysis, there were significances between these: the Kruskal–Wallis test, t = 9.8667, *p* = 0.0072, and then multiple comparison tests showed that there was significance between the 4 hAEL vs. the 8 hAEL and the 4 hAEL vs. the 12 hAEL (Table 1): a Tukey’s test, q = 7.79 in A/B, 9.0 in A/C, 1.25 in B/C, q = 5.91, α = 0.05, *k* = 3, *v* = 3.

### 2.2. Relationship Between the Developmental Stage of Eggs in Microinjection and Frequency of Transformation in Each Positive Brood (Cluster Size)

To investigate the relationships between the developmental stage of eggs used in microinjection and the frequency of transformation in each the DsRed2-positive brood, we analyzed the proportion of transformants per total embryos in a brood (cluster size) and distributed the positive broods in each cluster size. As shown in Figure 2, the transgenesis efficiency was observed widely in 0.3 to 58.8% at 4 hAEL, while it was observed to be in a smaller range of 2.0 to 9.2% at 8 hAEL.

### 2.3. Number of piggyBac Inserts Observed in Each Cluster

To assess relationships between the cluster size and number of *piggyBac* inserts in each DsRed2-positive brood, we performed Southern blot analysis using the probe shown in Figure 3A. We examined using two to five individuals of the 3rd instar per each the broods in G_1_. Southern blot analysis showed that more fragments tended to be observed at the 4 hAEL than that at 8 hAEL (Figure 3B). Surprisingly, the sample from the #3–6w brood derived from a cross with the host strain pnd-w1 contained a significantly higher number of inserts in comparison with the other samples derived from the sibling mating. In the latter case, both parents might likely have sources of transformed germplasm.

### 2.4. Transgenesis Using Eggs Stored for a Long Time at 10 °C

To extend the permissive period for microinjection in successful transgenesis, we conducted microinjections using eggs that had been stored at low temperatures for a long time. The eggs were collected an hour from the start of laying, aligned on a slide glass within another hour (2 hAEL), and then stored in a chamber of 10 °C for 5 or 24 h until microinjection, represented as 5 h_10 °C, and 24 h_10 °C, respectively (see Table 2). Each microinjection was completed within 30 min after their movement to a room temperature of 25 °C to keep the same experimental condition, and then the eggs were put in a moist box at 25 °C in the incubator until hatching. As shown in Table 2, transgenic silkworms could be produced in both low temperature conditions (5 h_10 °C and 24 h_10 °C). Surprisingly, these efficiencies were the same or even more than those of conventional experiment conditions (4 h_25 °C) not stored at low temperatures.

### 2.5. Knock-Out Experiment of BmBLOS2 Gene Using Eggs Stored at Low Temperatures

We examined transgenesis efficiency using eggs stored at 10 °C and then demonstrated that it was useful, as shown in Section 2.4. However, there is still the question of how the relationship between conditions of time and temperature should be set. The *3xP3-DsRed2* gene is a good marker gene to detect transgenic silkworms, but it is too time- and labor-consuming to see the results of transgenesis because the screening of transgenic silkworms must be performed in G_1_ after microinjection generation (G_0_). Additionally, the *3xP3-DsRed2* gene is not available to precisely detect transformation and genome modification events in G_0_. Therefore, to examine the relationship between temperature treatment and exposure time, we performed the knock-out experiments of the *BmBLOS2* gene, which can show the phenotype of oily skin, in larvae of G_0_ [16]. When we performed microinjection using the eggs stored at 25 °C for 4 h (the standard way) or 24 h, a strong *BmBLOS2* knock-out phenotype appeared in 4 h but not in 24 h (Figure 4b,c, Table 3). Next, we performed microinjection using the eggs stored at 10 °C for 24 h and 120 h. Consequently, we found the mutant phenotype in 24 h (Figure 4d, Table 3), but not in 120 h where the hatching was low (Figure 4e, Table 3). Each frequency of mutation appearance was 100% in 4 h_25 °C and 24 h_10 °C; however, it was demonstrated as being 0% in 24 h_25 °C and 120 h_10 °C (Table 3).

Additionally, we examined temperature treatment in detail using eggs stored at −20, −3, 0, 5, 10, 15, and 20 °C for 24 h. As a result, a strong oily mosaic phenotype was observed in the −3, 0, 5, and 10 °C experiments and a weak one in the 15 °C experiments (Appendix A). However, there was no hatching in the −20 °C experiment; the oily mosaic phenotype did not appear at 20 °C or in the “not injected” control (Appendix A). Appearance frequencies of each knock-out phenotype in the −3, 0, 5, 10, 15, and 20 °C experiments were 100% (13/13), 100% (7/7), 94.1% (16/17), 100% (26/26), 32.0% (8/25), and 0% (0/7), respectively, with the numbers in the parentheses representing the mutant vs. total (Table 4). Next, we examined the exposure time using eggs stored at 10 °C for 48, 72, 96, and 120 h and observed the appearance of the oily phenotype in the 4th instar larvae. The results show that the larvae in the 48 h treatment demonstrated a weak oily mosaic phenotype; however, this was not found in any of the other experiments of 72, 96, and 120 h (Appendix A). The frequency of the mutant phenotype in the 48, 72, 96, and 120 h treatments was 97.3% (36/37), 0% (0/36), 0% (0/28), and 0% (0/60), respectively, with the numbers in the parentheses representing the mutant vs. total (Table 4).

### 2.6. Evaluation of Mutation by High-Resolution Melting (HRM) Analysis

To evaluate the degree of breakage in the target region of *BmBLOS2*, we performed HRM analysis, which can detect mutations in amplified DNA sequences as changes in the melting curve. The genomic DNA was extracted as one batch of five larvae at the fourth instar in each experiment. In experiments where eggs were stored at different temperatures, wide and strong disruptions (red lines) were found in the results for the −3, 0, 5, and 10 °C conditions when compared to the standard curve representing the normal condition of 25 °C (black bold line). However, only minor disruptions were noted in the 15 °C condition, while none were noted in the 20 °C condition (Figure 5A). In the experiments examining the exposure time at 10 °C, there were weak disruptions at 48 h and minimal to no disruptions beyond 72 h. In contrast, there was a significant disruption noted at 24 h (Figure 5B). In conclusion, these results were consistent with those observed in the oily mosaic phenotype (Appendix A).

## 3. Discussion

It is generally accepted that embryonic microinjection on genome modification must be completed during a syncytial preblastodermal period [14,45,47,56,58,59]; therefore, we empirically carried out microinjection within the period of 4 to 8 hAEL. However, we still have not verified the influence of different stages of eggs. Our work in this study has made this empirical inference clear, and more concrete, which is critical for transgenesis efficiency; the production of silkworms was five times higher at 4 hAEL than at 8 hAEL, with no production observed at 12 hAEL (Table 1). Li et al. reported that, in a Cas9 knock-out experiment, microinjecting early-stage eggs with high concentrations of sgRNA and Cas9 mRNA was useful, resulting in both a high frequency and intensity of mutations, leading to severe phenotypic abnormalities [60]. Their report is consistent with our results and demonstrates that microinjection using eggs at an earlier stage may be useful for enhancing the efficiency of transgenesis.

Through analyzing the cluster size in each DsRed2-positive brood in G_1_, the brood at 4 hAEL existed in a broad range of at most 60%, while the ones at 8 hAEL concentrated in a range of less than 10% (Figure 2). Additionally, Southern blot analysis of positive individuals indicated that there were many more inserts at 4 hAEL than at 8 hAEL (Figure 3B). In fact, Pavlopoulos et al. found a similar result in the relationship between the cluster size and the number of insertions in an experiment comparing the DNA and RNA of *Minos* transposase for transformation. They found that RNA may more quickly supply its protein source of transposase than DNA [61]. Therefore, our data suggest that integration events into the genome may occur frequently in the early stages of eggs, which might have the possibility of drastically elevating the expression of targeted genes through multiple inserts.

In a previous study, Sudo et al. determined that the threshold temperature at which silkworm larvae can grow is 11 °C [57]. Based on their findings, we investigated the potential for maintaining suitable conditions for eggs used in microinjection. When we collected eggs within less than 2 hAEL, stored them in the chamber of 10 °C, and then performed their microinjection after 5 or 24 h, we were able to acquire the transformants even after 24 h. Surprisingly, the result at 24 h at 10 °C had a higher transgenesis efficiency than that stored at 4 h at 25 °C in a conventional manner (Table 2). In conclusion, we succeeded in maintaining a suitable condition for eggs for microinjection. Moreover, detailed research on the temperature and storage time of embryos in the *BmBLOS2* knock-out experiments showed that it might be possible to acquire transgenic or genome-modified silkworms when embryos are stored at low temperatures ranging from −3 to 10 °C for 24 h or even when using eggs stored at 10 °C until 48 h (Table 4, Figure 5). Unfortunately, we did not examine a lower temperature condition than 10 °C. Storage at a lower temperature than 10 °C might have the potential for successful genome modification in a longer period than the result at 48 h at 10 °C.

In addition, as useful information regarding the microinjection system, we note the new manipulator introduced in this study (Appendix A), which was semi-automatic, very precisely moving, and notably useful for microinjection users. This manipulator is particularly useful for those working with insects with hard eggshells, as it allows for a two-step process that could reduce the injection time without reducing transgenesis efficiency (Appendix A).

There are some insects that have a short embryonic development period of 1~3 days [45,47]; therefore, so far they must have collected a lot of eggs every given period, aligned eggs on to glass plates by many supporters, and completed microinjection in a short time. Although the threshold temperature should be found for each insect, our results may be very useful for researchers in similar situations, as they may make a time allowance for a microinjection and allow performing it alone.

## 4. Materials and Methods

### 4.1. Strain and Rearing

The pnd-w1 strain with the genotype of *pnd*/*pnd* and *w1*/*w1* [14] which are responsible for the non-diapause and the white egg color trait, respectively, was used as the host strain for microinjection. This strain was maintained at the Silkworm Research Group in the National Agriculture and Food Research Organization (NARO), Japan. The larvae were reared on a commercial diet including mulberry leaf powder, SilkMate PS (Nosan Corporation, Yokohama, Kanagawa, Japan), at a photoperiod of 12 h in the light and 12 h in the dark, at 28 °C for the 1st to the 4th and at 25 °C for the 5th instar larvae without controlling humidity, but around 40 to 60%.

### 4.2. Preparation of Plasmid DNA

The helper plasmid pHA3PIG [23] and the vector plasmid pBac(3xP3-DsRed2) [62] were used in this study. Both plasmid DNAs were extracted using a HiSpeed Plasmid Midi Kit (QIAGEN, Chuo, Tokyo, Japan) and solved in injection buffer including 0.5 mM phosphate buffer (pH 7.0) and 5 mM KCl, at a final concentration of 0.2 μg/μL each [14].

### 4.3. Preparation of Eggs

The eggs for the microinjection experiments were collected in less than an hour after oviposition according to the previous method [63]. They were aligned on slide glasses, fixed using commercial adhesive (JAN 4901490 305230, Konishi, Chuo, Osaka, Japan), and then employed for microinjection.

### 4.4. Microinjection

We introduced a new micro-manipulator, M300 (Micro Support Co., Tokyo, Japan, https://www.microsupport.co.jp/en/) in this study and designed an adjustor HD-21 (Entrusted company: Sankei Co., Koto, Tokyo, Japan). The manipulator memorizes the positions of a tungsten needle and a glass capillary and moves them using a computer-controlled electric motor, and the adjustor regulates their primary position (Appendix A). The previous system [63], the oil hydraulic manipulator (M152 and HDD-20, Narishige, Setagaya, Tokyo, Japan), was used as a comparison for the new system. The microinjected eggs were stored in a box with a humidity of 25 °C until hatching. Additionally, we used an electropolished tungsten needle (tip diameter: 5 μm, TP-005, Micro Support Co., Shizuoka, Shizuoka, Japan) which is very sharp and tolerant for many operations (Appendix A).

### 4.5. Screening of Transgenic Silkworms

The moths grew to adulthood after microinjection (G_0_) were single-mated among each other (sibling mating), and the remaining moths were mated with the host strain (pnd-w1). Then, the progeny eggs (G_1_) were collected from each female moth (brood). A fluorescence binocular microscope (SZX16, Olympus, Tokyo, Japan) with a DsRed filter (RFP1: SZX2-FRFP1, Olympus, Hachioji, Tokyo, Japan) was used to screen transgenic animals in the eggs.

### 4.6. Southern Blot Analysis

The genomic DNA samples were extracted from 3rd instar larvae in G_1_ and digested by the highly concentrated *Msp* I (100,000 units/mL, NEB, Sumida, Tokyo, Japan) for Southern blot analysis, performed according to the previous report [64]. The probe used to detect target fragments was amplified as a template of the right arm region of *piggyBac* inverted terminal repeat (ITR) by PCR [65] and labeled using the AlkPhos Direct Labelling and Detection System with CDP-Star (GE Healthcare Co., Hino, Tokyo, Japan). The detection of targets was performed using the chemiluminescence imaging system LAS-3000 (Fujifilm Co., Minato, Tokyo, Japan).

### 4.7. Genome Editing by the CRISPR-Cas9 System and Mutation Detection

The knock-out experiment of the *BmBLOS2* gene was designed according to Daimon et al. [16]. The single-guide RNA targeting the sequence “GAGTAGGGGTTGGATCTGCT” synthesized via the Alt-R^®^ CRISPR-Cas9 System (Integrated DNA Technologies, Coralville, IA, USA), and the Alt-R^®^ S.p. Cas9 Nuclease V3 (Integrated DNA Technologies, Coralville, IA, USA) were mixed with distilled water, resulting in final concentrations of 0.2 μg/μL each. The knockout phenotypes were observed using a microscopy (SZX16, Olympus, Tokyo, Japan) and photographed. The targeted sequence mutations were detected using high-resolution melting (HRM) analysis on the LightCycler^®^ 96 system (Roche, Basel, Switzerland), using KAPA HRM Fast PCR Kit (Roche, Basel, Switzerland), and the primer sets: “CGCGCATATTAAACGTTCCAG” and “TAGCTGTCGAGGCGGTAAGC”. The PCR was conducted with LightCycler^®^96 (Roche, Basel, Switzerland), using a two-step cycling protocol (50 cycles). The reaction contained 1× PCR Master Mix, 2.5 mM MgCl_2_, 0.25 μM of each primer, and 3 ng of w1-pnd genomic DNA in a total of 10 μL. The preincubation, denaturation, and annealing/extension conditions were set to 300 s at 95 °C, 10 s at 98 °C, and 30 s at 60 °C, respectively. The HRM analysis was conducted while gradually increasing the sample temperature from 65 °C to 97 °C (ramp: 0.04 °C/sec, sampling rate: 25 readings/°C) after 60 sec (40 °C) cooling step.

### 4.8. Statistical Analysis

All statistical analyses were performed using Microsoft^®^ Excel^®^ for Microsoft 365 MSO (Ver. 2402, Japan) or manually by calculation formula. The statistical analysis in Appendix A was performed using an unpaired *t* test, and the transgenic productivities between the conventional and new manipulators were compared. Multiple analysis in Table 1 was performed using a Kruskal–Wallis test and Tukey’s test to compare three parameters (4, 8, and 12 hAEL) on transgenic productivities. Significant variation was defined as *p* < 0.05.

## 5. Conclusions

We considered whether it was possible to sustainably maintain suitable conditions for embryonic microinjection to produce transgenic silkworms when time passed after egg laying. We found that appropriately cooling eggs significantly extended the permissive period for microinjection-mediated genome modification in *Bombyx mori*.

## 6. Patents

We obtained patents based on our research results in Japan; Patent No. P7475689.

## Figures and Tables

**Figure 1 ijms-25-12642-f001:**
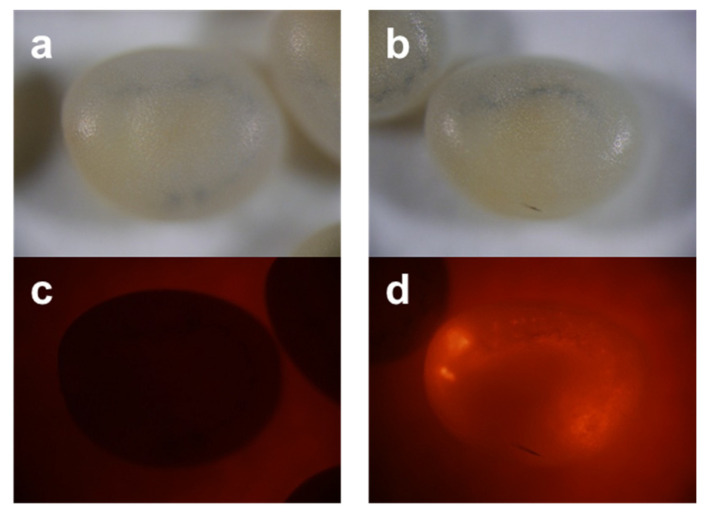
Photographs of the transgenic silkworm expressing DsRed2 driven by 3xP3 promoter. (**a**,**c**) are non-transgenic, and (**b**,**d**) are transgenic. (**a**,**b**) are in the bright field, and (**c**,**d**) are in the RFP filter.

**Figure 2 ijms-25-12642-f002:**
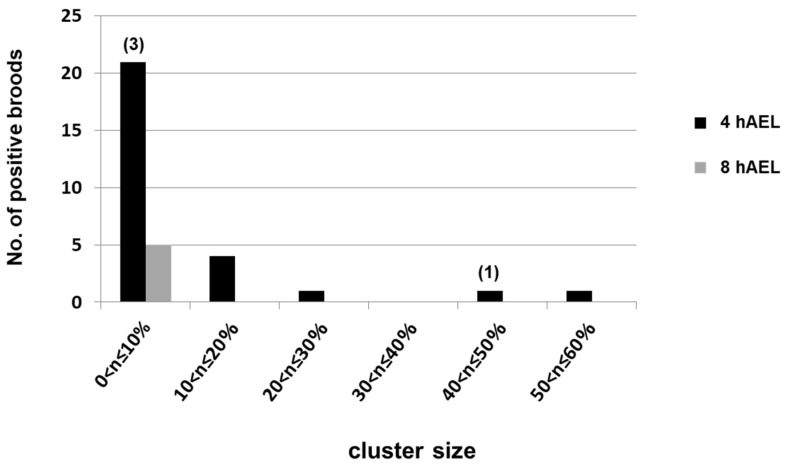
Distribution of cluster sizes (proportions of transformants per total embryos in each DsRed2-positive brood). The distributions here are counted as the total number of three experiments (#1–#3) shown in Table 1. The numbers in the brackets indicate the numbers of broods mated with the host strain moths pnd-w1; the others are from sibling mating.

**Figure 3 ijms-25-12642-f003:**
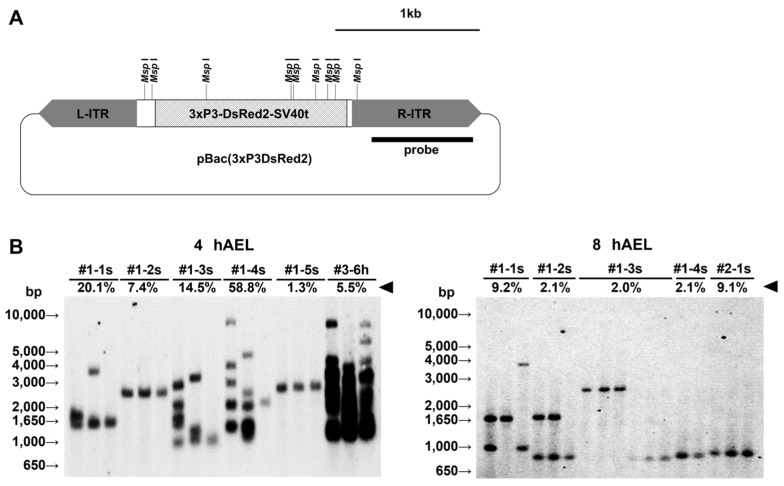
Southern blot analysis. (**A**) The illustration depicts the vector pBac(3xP3-DsRed2) in which the positions of *Msp* I cut sites and the probe used in the Southern blots are described. L- and R-ITR indicate an inverted terminal repeat of the left arm and the right arm, respectively. (**B**) Southern blot analysis in injection experiments of 4 and 8 hAEL. Each lane is derived from a different individual expressing DsRed2. The six samples of #1-3s at 8 hAEL are examined because there were two types with different marker expressions. The numbers starting with “#” indicate line names where the first is the experiment number, the second is the brood number, and the last letters “s” and “h” represent the manner of parental mating, where “s” stands for sibling mating, and “h” stands for mating with the host strain (pnd-w1), respectively. The arrowheads indicate the transformants’ proportion in each brood.

**Figure 4 ijms-25-12642-f004:**
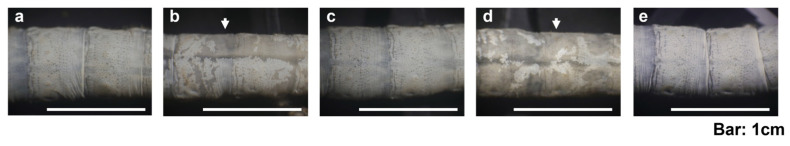
Photographs of the *BmBLOS2* knock-out phenotype at the 5th instar larvae. (**a**) not injected; (**b**) injected using the eggs stored at 25 °C for 4 h; (**c**) injected using the eggs stored at 25 °C for 24 h; (**d**) injected using the eggs stored at 10 °C for 24 h; (**e**) injected using the eggs stored at 10 °C for 120 h. The arrows at b and d indicate the oily mosaic phenotype.

**Figure 5 ijms-25-12642-f005:**
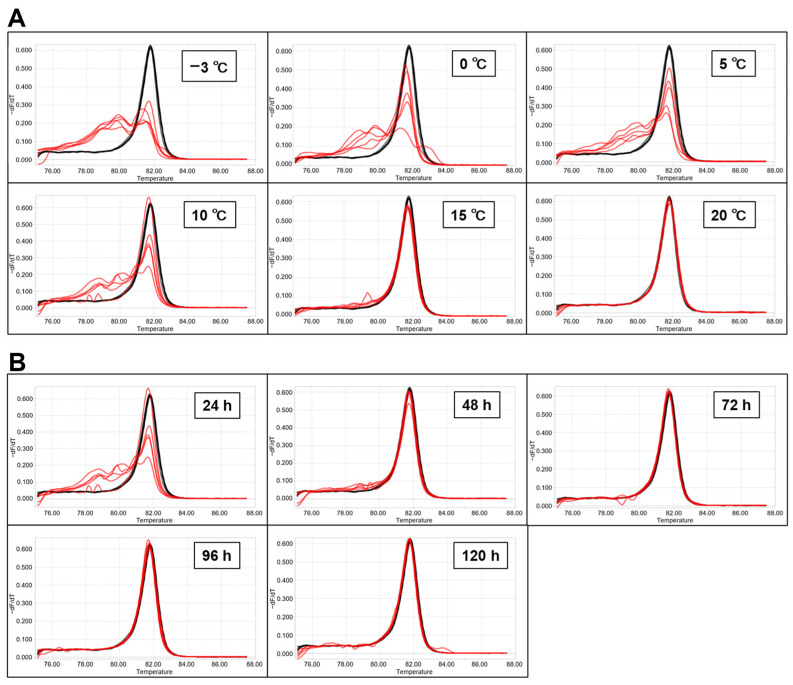
HRM analysis. (**A**) The analysis of eggs stored in different temperatures for 24 h. (**B**) The analysis of eggs stored at different times at 10 °C. The red lines are plotted from five extracted DNA samples derived from injected larvea, and the black lines represent the non-injected egg stored at 25 °C as the control. The vertical axis and the horizontal axis indicate differences in melting curves and temperature, respectively. All the DNA samples were extracted from five larvae of the 4th instar in G_0_. Both 10 °C data in (**A**) and 24 h in (**B**) are the same.

**Table 1 ijms-25-12642-t001:** Transgenesis efficiency using eggs that are differently elapsed.

Developmental Embryonic Stages	Experimental No.	(a) No. of Injected Eggs	(b) No. of Hatched Eggs (b/a×100)	Avg. Pct. of the column (b) (Mean ± SE)	No. of Fertile Adults	(c) Total No. of G_1_ Broods (s/h)	(d) Total No. of G_1_ DsRed2-Positive Broods (s/h)	(e) Pct of G_1_ DsRed2-Positive Broods (d/c×100)	Avg. Pct. of the column (e) (Mean ± SE)
4 hAEL	#1	236	98 (41.5%)		51	24 (19/5)	7 (7/0)	29.2%	
	#2	238	123 (51.7%)	53.6% ± 7.6%	70	30 (28/2)	11 (11/0)	36.7%	28.6% ± 4.8% a
	#3	237	160 (67.5%)		91	50 (30/20)	10 (6/4)	20.0%	
8 hAEL	#1	238	144 (60.5%)		87	45 (34/11)	4 (4/0)	8.9%	
	#2	240	143 (59.6%)	57.3% ± 2.8%	79	34 (30/4)	1 (1/0)	2.9%	1.7% ± 1.2% b
	#3	238	112 (47.1%)		54	33 (17/16)	0	0%	
12 hAEL	#1	240	82 (34.2%)		24	13 (8/5)	0	0%	
	#2	237	118 (49.8%)	38.1% ± 6.0%	56	27 (17/10)	0	0%	0.0% ± 0.0% b
	#3	238	72 (30.3%)		15	13 (2/11)	0	0%	
Non-injection	#1	250	137 (54.8%)		-	-	-	-	
	#2	250	232 (92.8%)	-	-	-	-	-	-
	#3	250	233 (93.2%)		-	-	-	-	

This experiment was performed to compare the efficiency of transgenic silkworm production using eggs at different developmental stages. The vector plasmid pBac(3xP3-DsRed2) and the helper plasmid pHAPIG were co-injected at a final concentration of 0.2 μg/μL into eggs at 4, 8, and 12 h at 25 °C: 4, 8, and 12 AEL. All G_0_ moths were crossed in a single-mating manner via sibling mating (s) or with the host strain: pnd-w1(h). Each experiment was performed in triplicate. (a)–(e) represent column symbol. The “Non-injection” status refers to a control for examining the hatchability of non-injected silkworms. The numbers in the parentheses indicate numbers of sibling mating and mating with the host strain: (s/h). a–b, significance between different signs.

**Table 2 ijms-25-12642-t002:** Transgenesis efficiency using the eggs stored at a low temperature.

Experiments	Treatment	Total No. of Examined Eggs	No. of Hatched Eggs	No. of Fertile Adults	Crossing Among the G_0_ Moths (Sibling Mating)		Crossing with Host Strain Moths	
					Total No. of G_1_ Broods	No. of G_1_ DsRed2-Positive Broods	Total No. of G_1_ Broods	No. of G_1_ DsRed2-Positive Broods
4 h_25 °C	Injected	217	121 (55.8%)	72	35	4 (11.4%)	2	0 (0%)
5 h_10 °C *	Injected	191	72 (37.7%)	50	25	3 (12.0%)	0	0 (0%)
24 h_10 °C *	Injected	162	121 (74.4%)	72	33	7 (21.2%)	6	0 (0%)
4 h_25 °C	Not injected	85	69 (81.2%)	-	-	-	-	-
24 h_10 °C *	Not injected	107	74 (69.2%)	-	-	-	-	-

This experiment was performed to compare transgenesis efficiency using eggs stored in different conditions of time and temperature. The experiments with “*” were performed using eggs stored at 10 °C for 5 or 24 h. The vector plasmid pBac(3xP3-DsRed2) and the helper plasmid pHAPIG were co-injected in a final concentration of 0.2 μg/μL. The experiment of “4 h_25 °C” was performed using eggs without a low temperature as the control; eggs of 4 hAEL at 25 °C in a conventional condition, were used. All G_0_ moths were crossed in a single manner of mating, either sibling mating or with the host strain, pnd-w1.

**Table 3 ijms-25-12642-t003:** Appearance rates of *BmBLOS2* knock-out phenotype in G_0_ using eggs stored at different times at 10 or 25 °C.

Experiments	Treatment	Total No. of Examined Eggs	No. of Hatched Eggs	No. of 5th Instar Larvae	No. of Positive Phenotypes
4 h_25 °C	Injected	90	25 (27.8%)	18	18 (100%)
	Not injected	174	92 (52.9%)	-	-
24 h_25 °C	Injected	91	36 (39.6%)	26	0 (0%)
24 h_10 °C	Injected	92	54 (58.7%)	49	49 (100%)
	Not injected	91	66 (72.5%)	-	-
120 h_10 °C	Injected	85	10 (11.8%)	9	0 (0%)
	Not injected	89	49 (55.1%)	-	-

The microinjections in “24 h_10 °C” and “120 h_10 °C” were performed using eggs stored at 10 °C for 24 and 120 h, respectively. The other experiments of “4 h_25 °C” and “24 h_25 °C” were performed using eggs without low temperature treatment as a control. The guide RNA, tracrRNA, and Cas9 nuclease were co-injected in a final concentration of 0.2 μg/μL.

**Table 4 ijms-25-12642-t004:** Appearance rates of the *BmBLOS2* knock-out phenotype in G_0_ using eggs stored at different temperatures and times.

Experiments	Treatment	Total No. of Examined Eggs	No. of Hatched Eggs	No. of 4th Instar Larvae	No. of Positive Phenotypes
−20 °C_24 h	Injected	96	0 (0%)	-	-
	Not injected	82	0 (0%)	-	-
−3 °C_24 h	Injected	96	22 (22.9%)	13	13 (100%)
	Not injected	77	47 (61.0%)	-	-
0 °C_24 h	Injected	96	9 (9.4%)	7	7 (100%)
	Not injected	82	48 (58.5%)	-	-
5 °C_24 h	Injected	96	27 (28.1%)	17	16 (94.1%)
	Not injected	66	48 (72.7%)	-	-
10 °C_24 h	Injected	96	34 (35.4%)	26	26 (100%)
	Not injected	64	44 (68.8%)	-	-
15 °C_24 h	Injected	96	31 (32.3%)	25	8 (32.0%)
	Not injected	77	61 (79.2%)	-	-
20 °C_24 h	Injected	96	10 (10.4%)	7	0 (0%)
	Not injected	71	59 (83.1%)	-	-
10 °C_48 h	Injected	96	41 (42.7%)	37	36 (97.3%)
	Not injected	72	55 (76.4%)	-	-
10 °C_72 h	Injected	96	45 (46.9%)	36	0 (0%)
	Not injected	66	57 (86.4%)	-	-
10 °C_96 h	Injected	96	45 (47.4%)	28	0 (0%)
	Not injected	65	50 (76.9%)	-	-
10 °C_120 h	Injected	96	63 (65.6%)	60	0 (0%)
	Not injected	70	40 (57.1%)	-	-

The source of the guide RNA, tracrRNA, Cas9 nuclease, and eggs for microinjection were prepared according to Table 3. In addition to the conditions in Table 3, several conditions, including temperatures (−20 °C to 20 °C) for 24 h and exposure times (for 24 to 120 h) at 10 °C, were examined. The mutant phenotype was observed at the 4th instar larvae.

## Data Availability

The datasets supporting the conclusions of this article are included within the article and its Appendix A.

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
