# Peer review of "Egg Cooling After Oviposition Extends the Permissive Period for Microinjection-Mediated Genome Modification in Bombyx mori"

_ijms, 2024, doi:10.3390/ijms252312642_

Round 1
Reviewer 1 Report
Comments and Suggestions for Authors
The manuscript presents a promising advancement in microinjection technology, offering improvements in the genetic modification process of insects with hard eggshells through the incorporation of a semi-automatic alignment system. This system shows considerable potential to reduce the time required for microinjections and enhance the productivity of genetic modifications, particularly in Bombyx mori. The study's findings on the effects of injection timing and egg refrigeration provide valuable insights into the field. While the overall structure of the paper is clear, several areas require significant improvement, especially in terms of experimental detail, statistical analysis, and methodology.
1. Statistical Analysis for Table 1 and Results Section 2.1: A statistical analysis is essential in Table 1 and Section 2.1 to compare the productive efficiency of transgenic silkworms using the conventional microinjection method versus the new microinjection manipulator. This comparison is critical to assess whether the new system offers a statistically significant improvement over traditional method. Including this analysis would strengthen the reliability of the study's conclusions.
2. Statistical Comparison of Efficiency Across Developmental Stages: Table 2 compares the productive efficiency of transgenic silkworms when microinjections are performed at different developmental stages. However, the table lacks statistical analysis to determine whether the differences observed at 4, 8, and 12 hours after egg laying (hAEL) are statistically significant. Implementing appropriate statistical tests would reinforce the validity of the study's conclusions regarding the impact of developmental timing on microinjection success.
3. Materials and Methods Section—Page 11: More details are required in the materials and methods section, particularly in the "Larval Preparation" subsection. Specific information on the commercial diet used, including its main components, nutritional content, and supplier details, should be provided. Additionally, the environmental conditions, such as photoperiod, temperature ranges, and humidity, under which the larvae were raised should be described in detail. These details are essential for replicating the study and understanding how environmental variables might influence experimental outcomes.
4. PCR Protocol Details—Page 12: The PCR conditions outlined on page 12 need further clarification. Specifically, details on the primers used (including their sequences and target regions for amplification), the thermal cycling conditions (denaturation, annealing, and extension temperatures and durations), and the concentrations of reagents (e.g., MgClâ‚‚, dNTPs, Taq polymerase) should be provided. This information is crucial for ensuring the reproducibility and accuracy of the genetic analysis performed in the study.
5. Statistical Methods Description: The manuscript lacks a clear explanation of the statistical methods used to analyze the data. It is important to include the type of statistical tests applied (e.g., ANOVA, t-tests, chi-square), the software used for analysis, and the criteria for statistical significance (e.g., p-values or confidence intervals). Including this section will provide transparency in the data analysis process and help readers understand how the results support the study’s conclusions.
While the manuscript presents valuable advancements in insect genetic modification through the development of an improved microinjection system, the findings appear more suitable for a patent application than for a research paper. The focus on the technical innovation of the semi-automatic alignment system and its practical benefits, such as increased efficiency and reduced time for microinjections, aligns closely with the objectives of a patent. To be appropriate for publication in a research journal, the study would benefit from deeper exploration of the underlying biological mechanisms, broader experimental validation, and more extensive analysis of the system's impact on genetic outcomes, beyond its technical performance.
Author Response
Comments 1:
The manuscript presents a promising advancement in microinjection technology, offering improvements in the genetic modification process of insects with hard eggshells through the incorporation of a semi-automatic alignment system. This system shows considerable potential to reduce the time required for microinjections and enhance the productivity of genetic modifications, particularly in Bombyx mori. The study's findings on the effects of injection timing and egg refrigeration provide valuable insights into the field. While the overall structure of the paper is clear, several areas require significant improvement, especially in terms of experimental detail, statistical analysis, and methodology.
Response 1:
Thank you for your polite review. Please consider the attached manuscript that has been revised in accordance with the reviews' comments. This version of manuscript was largely reconstructed in accordance with the Reviewers’ comments; we entirely described focusing an efficiency of cooling eggs for microinjection. Main revised places demanded from the reviewers were highlighted with yellow (for Reviewer 1), green (for Reviewer 2) and blue (for both reviewers) in the pdf file version (ijms-3258698_revision_1st marked revisions_241109: pdf file). Therefore, the title was replaced with “Egg cooling after oviposition extends the permissive period for microinjection-mediated genome modification in Bombyx mori.” Besides, we revised according to your following comments. Please confirm all in the revised manuscript (pdf file with color marks). Finally, the revised manuscript has already been edited English using the MDPI Author Services.
Comments 2:
Statistical Analysis for Table 1 and Results Section 2.1: A statistical analysis is essential in Table 1 and Section 2.1 to compare the productive efficiency of transgenic silkworms using the conventional microinjection method versus the new microinjection manipulator. This comparison is critical to assess whether the new system offers a statistically significant improvement over traditional method. Including this analysis would strengthen the reliability of the study's conclusions.
Response 2:
Thank you for your pointing out. We agree with this comment. We performed statistical analyses and added the results in the Table 1 which was changed as Table S1 as supplementary data according with another reviewers’ comments.
Comments 3:
Statistical Comparison of Efficiency Across Developmental Stages: Table 2 compares the productive efficiency of transgenic silkworms when microinjections are performed at different developmental stages. However, the table lacks statistical analysis to determine whether the differences observed at 4, 8, and 12 hours after egg laying (hAEL) are statistically significant. Implementing appropriate statistical tests would reinforce the validity of the study's conclusions regarding the impact of developmental timing on microinjection success.
Response 3:
Thank you for your pointing out. We agree with this comment. We performed statistical analyses and added the results in the Table 1 (previous Table 2) and in the line 76–88, as the corresponding text. Please see them in in the revised manuscript
Comments 4:
Materials and Methods Section—Page 11: More details are required in the materials and methods section, particularly in the "Larval Preparation" subsection. Specific information on the commercial diet used, including its main components, nutritional content, and supplier details, should be provided. Additionally, the environmental conditions, such as photoperiod, temperature ranges, and humidity, under which the larvae were raised should be described in detail. These details are essential for replicating the study and understanding how environmental variables might influence experimental outcomes.
Response 4:
Thank you for your pointing out. We agree with this comment. We added more detail information to the line 293–297. Please see the revised manuscript.
Comments 5:
PCR Protocol Details—Page 12: The PCR conditions outlined on page 12 need further clarification. Specifically, details on the primers used (including their sequences and target regions for amplification), the thermal cycling conditions (denaturation, annealing, and extension temperatures and durations), and the concentrations of reagents (e.g., MgClâ‚‚, dNTPs, Taq polymerase) should be provided. This information is crucial for ensuring the reproducibility and accuracy of the genetic analysis performed in the study.
Response 5:
Thank you for your pointing out. We agree with this comment. We added more information in the line 341–352. Please see the revised manuscript.
Comments 6:
Statistical Methods Description: The manuscript lacks a clear explanation of the statistical methods used to analyze the data. It is important to include the type of statistical tests applied (e.g., ANOVA, t-tests, chi-square), the software used for analysis, and the criteria for statistical significance (e.g., p-values or confidence intervals). Including this section will provide transparency in the data analysis process and help readers understand how the results support the study’s conclusions.
Response 6:
Thank you for your pointing out. We agree with this comment. We added more information in the line 353–359. Please see the revised manuscript.
Comments 7:
While the manuscript presents valuable advancements in insect genetic modification through the development of an improved microinjection system, the findings appear more suitable for a patent application than for a research paper. The focus on the technical innovation of the semi-automatic alignment system and its practical benefits, such as increased efficiency and reduced time for microinjections, aligns closely with the objectives of a patent. To be appropriate for publication in a research journal, the study would benefit from deeper exploration of the underlying biological mechanisms, broader experimental validation, and more extensive analysis of the system's impact on genetic outcomes, beyond its technical performance.
Response 7:
Thank you very much for your valuable comment. The manipulator of Micro Support is not our development excluding the device in the Figure S3D. We just introduced that to our study. We are not in relationship with the Micro Support besides of buyer and seller. We just present an information that the manipulator of Micro Support is very useful for microinjection users, especially users necessary the two-step microinjection because of hard eggshell. Therefore, we changed this paper's focus to an effect of cooling eggs for microinjection as main subject. Additionally, we revised in the line 309–311 of the “Materials and Methods” regarding a semi-automatic alignment system as following; “We introduced a new micro-manipulator, M300 (Micro Support Co., Tokyo, Japan, https://www.microsupport.co.jp/en/) in this study and designed an adjustor HD-21 (En-trusted company: Sankei Co., Tokyo, Japan).” We have already acquired the patent regarding the besides of manipulator and wrote about that in the line 366–367.
Reviewer 2 Report
Comments and Suggestions for Authors
This paper describes a modified microinjection system to enable GM of insects with hard eggshells, and also the use of refrigeration to extend the window when microinjection is possible.
The abstract shifts from describing a a semi-automatic alignment system to the impact of refrigeration, so it's not entirely clear what the focus of this paper is / what the innovation or modification the authors have invented is. I would like to see this stated more clealry in the abstract. Say, for example: "an improved microinjection technology featuring a semi-automatic alignment system for the capillary with the hole, significantly reducing injection time, combined with an optional refrigeration step to prolong the syncytial preblastodermal stage and extend the time window in which transgenesis is possible." Note that, if you did not invent this new system yourself, then you cannot take credit for it, and your paper should focus on the refrigeration.
The majority of this paper focuses on the effect of refigeration, but the title focuses on the modified system. Change the title to reflect the true focus of this paper. Again, if you did not invent the modified system, but just bought the pre-made system from Micro Support, then change the paper to completely focus on the refrigeration part.
The introduction is too short, in part because much of the appropriate text is in the discussion. The Discussion section first discusses the results, then explains what they means for the field, and finally what the future research should look like. Any information that provides background to why you did this study belongs in the introduction only. Lines 231-247 of the disucssion must thus be moved, almost unchanged, to the introduction. Start the Discussion section with "Here we introduced…"
For this paper, I strongly recommend placing the methods before the results unless the journal forbids it.
It's not clear in the methods or results what is the creation of the authors and what is the creation of the company Micro Support. Are you testing their new invention, or did you invent it? If you work for Micro Support, then you need to disclose this in your affiliations and also in the Conflicts of Interest. As written, I do not see what the authors did other than compare the M300 with the M152 that somebody else invented and add refrigeration.
It is not clear from the methods what you micro-injected into the eggs. What gene were you trying to modify? DsRed2, I assume, but why this gene? What does it do? And later, why did you knock-out BmBLOS2? What does it do, and why is the knock-out worth doing?
The success rate of microinjection here needs to also be comapred to those published in other studies, as the numbers appear very low. Is this normal?
Other comments
34-35 Unclear. You made a 2-step system, but the conventional method (one-step system?) remains a hurdle for which kind of microinjection users? There seem to be two unrelated ideas here.
49 "size" is not the right word, unless you are describing the thickness of the needle and capillary. Did you mean distance?
67 hAEL needs to be defined again here. The abstract doesn't count.
115 Perhaps it is because this is not my field, but I do not understand what "cluster size" is, or what this Table is supposed to indicate.
119 sib means "sibling," yes? Say sibling.
123 Delete "basically"
127 Delete "terribly"
132 "long time" is a bold claim. In the introduction you would need to proviude evidence that microinjection is not typically done on eggs stored for so long. What is the current state of research of microinjection on refrigerated insect eggs?
155-161 For the eggs stored at 10°C, were they placed in storage immediately after egg laying, or 4hAEL?
191 Something is wrong with that dash in "c-g". It's a different font than the rest of the paper. Replace it.
Figure 6: the font is too small on the axes. I can't read the graphs.
247-248 Did you introduce the new system, or did Micro Support introduce it and you just used it?
251 New information (table S1) cannot be mentioned in the discussion. You should have this information in the results section.
252 "capitally" is the wrong word
253 replace "additionally important as well" with "also important"
254-255 This is the first time in the paper where you actually explained what you did: you changed a physically polished needle [of unknwon material] with a tungsten one. That needs to be in the methods.
257 Again, Figure S2 needs to be cited first in the Results, not the Discussion. It also sounds important enough to be a normal figure.
261 "we" is unclear: do you mean you the authors, or we the scientific community? Rephase this sentence.
262-263 "obvious, and more concretely, which is amazingly critical" these words are overly strong and unprofessional sounding. State in neutral, boring terms what you actually did. The reader will decide if this is important or not.
265-270 It sounds like it was already very well known that ealier microinjection leads to higher success. Your research and Table 2 thus contribute little to science. That's why "Our work in this study has made this empirical inference obvious, and more concretely, which is amazingly critical in producing transgenic silkworm" is so unprofessional sounding. From my reading, you didn't discover anything new.
286 "Conclusionary" is not the right word. Say "in conclusion" or "thus"
291 "Our refined system" is not clear. I do not think your phsyical microinjection system is necessary for refrigeration to work. Again, this paper seems to be about two seperate subjects: a new microinjection system and the impact of refrigeration. You need to delete every single overly-dramatic claim about how great your results are, and be very clear about what you actually did.
294 "our research on microinjection technology is still on the way. Ultimately, we hope to build a full-automatic microinjection system." I do not believe for one second, based on this sentence, that none of the authors have a financial conflict of interest. At least of you is paid or employed by Micro Support. Be honest! Else, if you really are independent, be more clear about what this paper is about [refrigeration], and talk as little as possible about a "new" micro-injection system invented by a company you do not work for.
354 "dramatic leap in time and ease of operation" is a bold claim, but you didn't provide any evidence for it in the paper. If this is an important part of your conclusions, then change Supplementary Table 1 to a normal Table, or at least make a publication-worthy version of the data.
Some English editing by a native speaker is needed.
Author Response
Comments 1:
This paper describes a modified microinjection system to enable GM of insects with hard eggshells, and also the use of refrigeration to extend the window when microinjection is possible.
Response 1:
Thank you for your polite review. Please consider the attached manuscript that has been revised in accordance with the reviews' comments. This version of manuscript was largely reconstructed in accordance with the Reviewers’ comments; we entirely described focusing an efficiency of cooling eggs for microinjection. Main revised places demanded from the reviewers were highlighted with yellow (for Reviewer 1), green (for Reviewer 2) and blue (for both reviewers) in the pdf file version (ijms-3258698_revision_1st marked revisions_241109: pdf file). Therefore, the title was replaced with “Egg cooling after oviposition extends the permissive period for microinjection-mediated genome modification in Bombyx mori.” Besides, we revised according to your following comments. Please confirm all in the revised manuscript (pdf file with color marks). Finally, the revised manuscript has already been edited English using the MDPI Author Services. As I repeatedly mention below, I sincerely response and never lie regarding with this paper. Please neutrally review this paper.
Comments 2:
The abstract shifts from describing a a semi-automatic alignment system to the impact of refrigeration, so it's not entirely clear what the focus of this paper is / what the innovation or modification the authors have invented is. I would like to see this stated more clealry in the abstract. Say, for example: "an improved microinjection technology featuring a semi-automatic alignment system for the capillary with the hole, significantly reducing injection time, combined with an optional refrigeration step to prolong the syncytial preblastodermal stage and extend the time window in which transgenesis is possible." Note that, if you did not invent this new system yourself, then you cannot take credit for it, and your paper should focus on the refrigeration.
Response 2:
Thank you very much for pointing out of the “Abstract.” We agree with this comment. In accordance with your suggestion, I entirely reconstructed the abstract with an effect of refrigeration as the main subject. Please confirm the “Abstract” of the revised manuscript.
Comments 3:
The majority of this paper focuses on the effect of refigeration, but the title focuses on the modified system. Change the title to reflect the true focus of this paper. Again, if you did not invent the modified system, but just bought the pre-made system from Micro Support, then change the paper to completely focus on the refrigeration part.
Response 3:
Thank you for your appropriate comment. We accepted your comment. The title was replaced with “Egg cooling after oviposition extends the permissive period for microinjection-mediated genome modification in Bombyx mori.” Please confirm the revised manuscript. In accordance with your suggestion, I entirely reconstructed this paper to an effect of cooling eggs as main subject. Please confirm the revised manuscript.
Comments 4:
The introduction is too short, in part because much of the appropriate text is in the discussion. The Discussion section first discusses the results, then explains what they means for the field, and finally what the future research should look like. Any information that provides background to why you did this study belongs in the introduction only. Lines 231-247 of the disucssion must thus be moved, almost unchanged, to the introduction. Start the Discussion section with "Here we introduced…"
Response 4:
Thank you for your pointing out. We agree with this comment. We reconstructed the “Introduction” and “Discussion.” The early part of the “Discussion” of the previous version was moved to the “Introduction” in accordance with your suggestion. Please see blue marked place in the “Introduction.”
Comments 5:
For this paper, I strongly recommend placing the methods before the results unless the journal forbids it.
Response 5:
I am sorry. The format is set in advance in the IJMS; the Materials and Methods must follow the Results.
Comments 6:
It's not clear in the methods or results what is the creation of the authors and what is the creation of the company Micro Support. Are you testing their new invention, or did you invent it? If you work for Micro Support, then you need to disclose this in your affiliations and also in the Conflicts of Interest. As written, I do not see what the authors did other than compare the M300 with the M152 that somebody else invented and add refrigeration.
Response 6:
Thank you for your comment. The manipulator of Micro Support is not our development excluding the device in the Figure S3D. As you say, we just introduced that to our study. We are not in relationship with the Micro Support besides of buyer and seller. We just presented an information that the manipulator of Micro Support was very useful for microinjection users, especially users necessary the two-step microinjection because of hard eggshell. Therefore, we changed this paper's focus to an effect of cooling eggs as the main subject. Additionally, the description of the semi-automatic manipulator system was rewritten in the line 309–311 as following; the “We introduced a new micro-manipulator, M300 (Micro Support Co., Tokyo, Japan, https://www.microsupport.co.jp/en/) in this study and designed an adjustor HD-21 (En-trusted company: Sankei Co., Tokyo, Japan).”
Comments 7:
It is not clear from the methods what you micro-injected into the eggs. What gene were you trying to modify? DsRed2, I assume, but why this gene? What does it do? And later, why did you knock-out BmBLOS2? What does it do, and why is the knock-out worth doing?
Response 7:
Thank you for your comment and question. We understood that your comment means because of lack of our explanation. The DsRed2 gene driven by 3xP3 promoter (3xP3-DsRed2) was used as a marker gene to detect transgenic silkworms, not to modify somewhat gene in the genome. The 3xP3-DsRed2 gene is a good marker gene to detect transgenic silkworm but that takes time to see the results; the screening of transgenic silkworm must be done in G1: next generation of microinjection (G0). Additionally, the 3xP3-DsRed2 gene is not available to preciously detect transformation and genome modification events in G0. Therefore, we chose the knock-out experiment of BmBLOS2 in the egg-cooling experiments. As shown in the Figure 4, BmBLOS2 knock-out can appear the mutant phenotype, oily skin, in the G0 larvae, which should reflect an event breaking the target sequence in the genome at the early egg stage during microinjection. Moreover, we could detect the event in the genome by HRM analysis even in G0. We added the above explanation in the line 162–171. Please see the revised manuscript.
Comments 8:
The success rate of microinjection here needs to also be comapred to those published in other studies, as the numbers appear very low. Is this normal?
Response 8:
Thank you for your comment and question. The transgenesis frequency in this study is not low. This is normal in the study using Bombyx mori because the success rate was 1.6–26.7% in the paper Dr. Tamura reported excluding that those days they must injected some thousands of eggs (see the Reference no. 14) The success rate is dependent on the type of vector, implementers skill, and various other factors. We performed this study using one vector and being injected by one person to estimate in the same condition as possible as. Therefore, although the comparison with the other studies might has some sense, we believe that the concept and the method in this study is better. However, since it was pointed out to us, I have added the efficiencies from previous studies in the line 54–56. Please see the revised manuscript.
Comments 9:
34-35 Unclear. You made a 2-step system, but the conventional method (one-step system?) remains a hurdle for which kind of microinjection users? There seem to be two unrelated ideas here.
Response 9:
Thank you for your question. The conventional method (manipulator) in this paper is also two-step system. The two-step microinjection is significantly difficult comparing to the one-step as used in mammal or other animals. In the case of the conventional type, the alignment of a needle and capillary have been inaccurate, so users, especially for novice researchers, had to change capillaries several time because of breaking of it during microinjection. Therefore, that takes time loss and then the suitable timing for microinjection will pass. However, we removed most of description regarding new manipulator system from the revised manuscript because we changed the main subject of this paper to an effect of cooling eggs. Please confirm the revised manuscript.
Comments 10:
49 "size" is not the right word, unless you are describing the thickness of the needle and capillary. Did you mean distance?
Response 10:
Thank you for your pointing out. However, we removed most of description regarding new manipulator system from the revised manuscript because of the above. Therefore, that corresponding sentence including the "size" have already been removed. Please confirm the revised manuscript.
Comments 11:
67 hAEL needs to be defined again here. The abstract doesn't count.
Response 11:
Thank you for your pointing out. We agree with this comment. We revised as you say. Please confirm in the line 74.
Comments 12:
115 Perhaps it is because this is not my field, but I do not understand what "cluster size" is, or what this Table is supposed to indicate.
Response 12:
Thank you for your comment and question on the "cluster size." W received that your question is regarding to the Figure 2 (previous Figure 3), not "Table." The cluster size was presented by Pavlopoulos et al., which was defined as the "percentage of transformed G1’s obtained per G0" (reference no. 64). The Figure 2 represents the distribution of the cluster size between the 4 and 8 hAEL in the microinjection. You can see that the 4 hAEL's cluster size is seen in the higher range than that of the 8 hAEL; the DsRed2 positive brood in the higher percentage cluster size assumed to have more multiple inserts than that of the lower ones. Therefore, we performed the Southern Blot analysis to confirm the multiple insertion. We added more explanation about these in the line 107–110 and the line 118–120. Please see in the revised manuscript.
Comments 13:
sib means "sibling," yes? Say sibling.
Response 13:
Thank you for pointing out of the “sib.” We replaced it with “sibling” in the whole text. Please see revised places marked yellow in the revised manuscript.
Comments 14:
123 Delete "basically"
Response 14:
Thank you for your suggestion. We accepted your comment and deleted the "basically" in the line 120 as the “We examined…...”
Comments 15:
127 Delete "terribly"
Response 15:
Thank you for your suggestion. The corresponding word was removed.
Comments 16:
132 "long time" is a bold claim. In the introduction you would need to proviude evidence that microinjection is not typically done on eggs stored for so long. What is the current state of research of microinjection on refrigerated insect eggs?
Response 16:
Thank you for your suggestion. In some insects with short embryonic development such as Drosophila, Caenorhabditis, and so on, microinjection was performed within some minutes to a few hours in general (see reference no 45, 46). We add description regarding your question in the 47–49 of the “Introduction.” Please see in the revised manuscript.
Comments 17:
155-161 For the eggs stored at 10°C, were they placed in storage immediately after egg laying, or 4hAEL?
Response 17:
Thank you for your question. That is 2 hours after egg laying.
Comments 18:
191 Something is wrong with that dash in "c-g". It's a different font than the rest of the paper. Replace it.
Response 18:
Thank you for your pointing out. I checked and got corrected in the line 197.
Comments 19:
Figure 6: the font is too small on the axes. I can't read the graphs.
Response 19:
Thank you for your pointing out.
As you say, it is sure that the axes information is small. We made efforts to expand those, but it is difficult to expand enough because of the data from PCR device. However, the importance of data is a fluctuation of the red lines, so the data serves its purpose.
Comments 20:
247-248 Did you introduce the new system, or did Micro Support introduce it and you just used it?
Response 20:
Thank you for your comment. I answered your question in the “Response 6.” Please see it.
Comments 21:
251 New information (table S1) cannot be mentioned in the discussion. You should have this information in the results section.
Response 21:
Thank you for your suggestion. We changed Table S1 as Table S2. We added the information in the line 279.
Comments 22:
252 "capitally" is the wrong word
Response 22:
Thank you for your pointing out. That was removed in the process of the revision. We can’t find it in the new manuscript.
Comments 23:
253 replace "additionally important as well" with "also important"
Response 23:
Thank you for your pointing out. That was removed in the process of the revision. We can’t find it in the new manuscript.
Comments 24:
254-255 This is the first time in the paper where you actually explained what you did: you changed a physically polished needle [of unknwon material] with a tungsten one. That needs to be in the methods.
Response 24:
Thank you for your comment. That new tungsten needle was written in the “Methods” (line 316).
Comments 25:
257 Again, Figure S2 needs to be cited first in the Results, not the Discussion. It also sounds important enough to be a normal figure.
Response 25:
Thank you for your comment. We reconstructed this paper with an effect of cooling eggs for microinjection as main subject in accordance with your comments, so we think that Figure S2 is not important as it was. Therefore, that may be not necessary.
Comments 26:
261 "we" is unclear: do you mean you the authors, or we the scientific community? Rephase this sentence.
Response 26:
Thank you for your comment. We mean our authors’ group but have ever not heard that other researchers examined. We don’t feel this sentence issue. Please let us know your meaning in detail.
Comments 27:
262-263 "obvious, and more concretely, which is amazingly critical" these words are overly strong and unprofessional sounding. State in neutral, boring terms what you actually did. The reader will decide if this is important or not.
Response 27:
Thank you for your comment. We revised this sentence to mild one. Please confirm the line 240–241. However, we inform you that there is a significant difference in statistical analysis regarding with that data.
Comments 28:
265-270 It sounds like it was already very well known that ealier microinjection leads to higher success. Your research and Table 2 thus contribute little to science. That's why "Our work in this study has made this empirical inference obvious, and more concretely, which is amazingly critical in producing transgenic silkworm" is so unprofessional sounding. From my reading, you didn't discover anything new.
Response 28:
Thank you very much for your comment. However, you are making a mistake. Because the description of the line 265-270 is a study by Li et al. who studied using the insects, Papilio Xuthus and Papilio machaon, not Bombyx mori. And their study is regarding with gene functional analysis which isn’t a study of an efficiency of transgenesis. In their paper there is a short description of efficiency of embryogenesis (they use CRISPR/Cas9) as following; “we conclude that the key factors for successful gene editing in butterflies include high concentrations of sgRNA and Cas9 mRNA, an appropriate ratio, mixed injection of two or more sgRNAs with close targeting sites and timing egg injection to target early embryogenesis.” This is their sole description regarding with microinjection in early embryonic stage. But they never show the evidence of efficiencies of microinjection in the early period of eggs. It is not true that you mention to be already known with scientific proof. Therefore, our data is the first evidence regarding data of microinjection during early stage and an efficiency of cooling eggs for microinjection. Please confirm their paper (reference no. 60). Regarding Table 1 (previous Table 2), the data shown in the Table 1 is very important to examine how long it is possible to produce transgenic animals during a normal development because with a criterion of data, basic data as Table 1, we can examine a possibility of permissive period of eggs for microinjection.
Comments 29:
286 "Conclusionary" is not the right word. Say "in conclusion" or "thus"
Response 29:
Thank you for your pointing out. We changed that word to the “in conclusion.” Thanks again.
Comments 30:
291 "Our refined system" is not clear. I do not think your phsyical microinjection system is necessary for refrigeration to work. Again, this paper seems to be about two seperate subjects: a new microinjection system and the impact of refrigeration. You need to delete every single overly-dramatic claim about how great your results are, and be very clear about what you actually did.
Response 30:
Thank you very much for your appropriate comment and I am sorry for confusing you and readers. We reconstructed this paper with focus in an effective of cooling of eggs for microinjection, so the previous description regarding with new microinjection system was reduced at minimum. Please confirm the revised manuscript.
Comments 31:
294 "our research on microinjection technology is still on the way. Ultimately, we hope to build a full-automatic microinjection system." I do not believe for one second, based on this sentence, that none of the authors have a financial conflict of interest. At least of you is paid or employed by Micro Support. Be honest! Else, if you really are independent, be more clear about what this paper is about [refrigeration], and talk as little as possible about a "new" micro-injection system invented by a company you do not work for.
Response 31:
Sincerely thank you for your comment. You can’t already find the corresponding sentence in the revised manuscript. I apologize that we describe regarding with the new manipulator at more than necessity. There is no financial conflict of interest between Micro Support and us. As mentioned in the “Response 6” it is just that we/(I) are simply moved the goodness of the device and because this is first time to inform the device in official paper. Nothing more, nothing less. Therefore, the previous description regarding with the new microinjection system was reduced at minimum. Please confirm the revised manuscript.
Comments 32:
354 "dramatic leap in time and ease of operation" is a bold claim, but you didn't provide any evidence for it in the paper. If this is an important part of your conclusions, then change Supplementary Table 1 to a normal Table, or at least make a publication-worthy version of the data.
Response 32:
Thank you for your pointing out. We repeatedly mention that we have already revised the new version of manuscript focusing on an efficiency of cooling eggs for microinjection. There is little description regarding with the new manipulator in the revised manuscript excluding the “Materials and Methods.” Please confirm the revised manuscript.
Round 2
Reviewer 1 Report
Comments and Suggestions for Authors
The revised manuscript has been sufficiently improved to merit publication in IJMS.
Reviewer 2 Report
Comments and Suggestions for Authors
The authors have accepted most of my recommendations and changed the paper's focus appropriately. I am satisfied with this new version, and recommend acceptance. Thank you!